# MRD-Based Therapeutic Decisions in Genetically Defined Subsets of Adolescents and Young Adult Philadelphia-Negative ALL

**DOI:** 10.3390/cancers13092108

**Published:** 2021-04-27

**Authors:** Manuela Tosi, Orietta Spinelli, Matteo Leoncin, Roberta Cavagna, Chiara Pavoni, Federico Lussana, Tamara Intermesoli, Luca Frison, Giulia Perali, Francesca Carobolante, Piera Viero, Cristina Skert, Alessandro Rambaldi, Renato Bassan

**Affiliations:** 1Hematology Unit, Azienda Socio Sanitaria Territoriale (ASST), Ospedale Papa Giovanni XXIII, 24127 Bergamo, Italy; mtosi@asst-pg23.it (M.T.); ospinelli@asst-pg23.it (O.S.); robertacavagna@outlook.com (R.C.); cpavoni@asst-pg23.it (C.P.); flussana@asst-pg23.it (F.L.); tintermesoli@asst-pg23.it (T.I.); arambaldi@asst-pg23.it (A.R.); 2Hematology Unit, Azienda Ulss3 Serenissima, Ospedale dell’Angelo, 30174 Venezia-Mestre, Italy; matteo.leoncin@aulss3.veneto.it (M.L.); luca.frison@aulss3.veneto.it (L.F.); giulia.perali@aulss3.veneto.it (G.P.); francesca.carobolante@aulss3.veneto.it (F.C.); piera.viero@aulss3.veneto.it (P.V.); cristina.skert@aulss3.veneto.it (C.S.); 3Department of Oncology-Hematology, University of Milan, 20122 Milan, Italy

**Keywords:** acute lymphoblastic leukemia, adolescents and young adults, risk stratification, minimal residual disease, risk-oriented therapy

## Abstract

**Simple Summary:**

In acute lymphoblastic leukemia (ALL), once a complete remission is achieved following induction chemotherapy, the study of submicroscopic minimal residual disease (MRD) represents a highly sensitive tool to assess the efficacy of early chemotherapy courses and predict outcome. Because of the significant therapeutic progress occurred in adolescent and young adult (AYA) ALL, the importance of MRD in this peculiar age setting has grown considerably, to refine individual prognostic scores within different genetic subsets and support specific risk and MRD-oriented programs. The evidence coming from the most recent MRD-based studies and the new therapeutic directions for AYA ALL are critically reviewed according to ALL subset and risk category.

**Abstract:**

In many clinical studies published over the past 20 years, adolescents and young adults (AYA) with Philadelphia chromosome negative acute lymphoblastic leukemia (Ph− ALL) were considered as a rather homogeneous clinico-prognostic group of patients suitable to receive intensive pediatric-like regimens with an improved outcome compared with the use of traditional adult ALL protocols. The AYA group was defined in most studies by an age range of 18–40 years, with some exceptions (up to 45 years). The experience collected in pediatric ALL with the study of post-induction minimal residual disease (MRD) was rapidly duplicated in AYA ALL, making MRD a widely accepted key factor for risk stratification and risk-oriented therapy with or without allogeneic stem cell transplantation and experimental new drugs for patients with MRD detectable after highly intensive chemotherapy. This combined strategy has resulted in long-term survival rates of AYA patients of 60–80%. The present review examines the evidence for MRD-guided therapies in AYA’s Ph− ALL, provides a critical appraisal of current treatment pitfalls and illustrates the ways of achieving further therapeutic improvement according to the massive knowledge recently generated in the field of ALL biology and MRD/risk/subset-specific therapy

## 1. Introduction

Acute lymphoblastic leukemia (ALL) is the most common type of cancer in children, peaking at 7.6:100.000/year at 1–4 years of age and decreasing in adolescents and young adults (AYA), to an average incidence of 1.8 and 0.8 between 15 and 39 years of age [1,2]. Therapeutic progress has been outstanding in childhood ALL over the past decades, now reaching an 80–90% chance of cure [3,4]. Although therapeutic results are progressively less favorable with the increase of patient age, the case of AYA patients is unique. AYAs have often been treated by adult ALL specialists with adult ALL protocols, a setting in which the favorable prognostic influx of a younger patient age has been consistently recognized [5]. Moreover, AYA patients share several diagnostic and clinical features in common with children and exhibit a higher tolerance to modern highly effective pediatric regimens, which makes them eligible to receive these pediatric-type treatments. The latter concept was explored in several clinical trials that eventually confirmed and provided an explanation for the therapeutic superiority of modern pediatric regimens compared to standard adult protocols in AYA ALL [6].

In ALL, beyond treatment itself that plays by definition a primary prognostic role, among all recognizable risk factors, a key point concerns the course of minimal residual disease (MRD) during the early consolidation phase in patients who respond well and achieve a complete hematologic remission (CR) following induction chemotherapy [7,8]. MRD is a fundamental, independent prognostic parameter that reflects the dynamics of chemo-sensitivity of the residual submicroscopic ALL cell burden to a given chemotherapy program in individual patients, whatever the underlying disease subset, ALL genetics and clinical risk profile. For these reasons, the collection of MRD data has become an essential component of modern treatment strategies for ALL at all ages, first to optimize the definition of risk class and provide an individual prognosis and, second, of equal or even greater importance, to align treatment type and intensity with the MRD-related risk of systemic relapse [9,10].

In the present review, we examine the evidence for the use of MRD analysis in the upfront management of AYA ALL. Because until now MRD-oriented trials were almost exclusively performed in Philadelphia chromosome-negative (Ph−) ALL, we exclude from our survey the subset of Ph+ ALL, which is also significantly less frequent in AYAs compared to older patients. We wish to underline that, once successfully treated, AYA patients, including children, are those who may experience the greatest benefit in terms of future life years and quality of their life. Disclosing the potential for cure or on the contrary the risk of failure through an MRD analysis integrated with other significant risk parameters is therefore of paramount importance in AYAs, to support and optimize risk-related treatment choices and results.

## 2. ALL in Adolescents and Young Adult Patients

Quite frequently, in the past, AYA patients with Ph− ALL have been included in adult ALL trials enrolling patients within a broad age range (from 15–18 to 55–65 years). While it may be difficult to extrapolate exact AYA data and results from these trials, they remain an important source of information regarding AYA ALL since the median patient age in most adult ALL studies is around 35–40 years and therefore some 50% of study patients fall within the AYA category. Instead, selected age ranges were rather heterogeneous in dedicated AYA trials, variably extending up to between 25 and 45 years, which makes it similarly difficult, as well as somewhat artificial, to set exact and universal age boundaries for AYA ALL. In addition, in some large pediatric trials, the upper patient age was extended to include all teens and younger adults, up to 25–30 years [11,12,13] or even 45 years [14], sometimes without clearly separating outcome results of children from AYAs [12,13]. In all these studies, the diagnostic characteristics of AYA ALL, genetics in first place, were intermediate between those observed in children and older adults. Overall, once the challenge of improving the outcome of AYA patients was correctly perceived and the first innovative trials demonstrated the advantages of using modern pediatric rather than historical adult programs, the new approach was rapidly adopted by virtually all cooperative adult ALL Study Groups worldwide [9,15,16,17] and is highly recommended at present. All these topics are considered in the following sections.

### 2.1. AYA Patient Identification: Age Ranges

While AYAs can represent roughly one half of all study patients in adult ALL trials (median patient age 35–40 years), there is no formal consensus for the patient age range in AYA-dedicated trials, this ranging from 15–18 to 35–45 years. Here, we refer for simplicity to AYA ALL as 18–40 years of age, with exceptions that are reported. It is worth recognizing that even patients aged 40–45 to 50–55 years were rather successfully treated with the same pediatric-based regimens used in children and AYAs.

### 2.2. Ph− ALL: Diagnostic Subsets Defined by Immunophenotype and Cytogenetics/Genetics in AYAs

Whereas the incidence of the two main immunophenotypic ALL subsets does not vary significantly across age groups, with B-cell precursor (BCP) ALL representing the majority of new cases and T-ALL no more than 20–25%, the distribution of different genetic subsets shows a definite age-related pattern, with a lower incidence of the favorable ones in AYAs compared to children (Table 1).

The majority of chromosomal and genetic alterations detectable in AYA ALL belong to the intermediate risk (including the normal karyotype subset) or intermediate-high risk categories, or even high-risk categories [18,19,20,21], such as the Ph-like ALL variant, which is a prognostically adverse entity. Genetically, Ph-like ALL resembles in many ways Ph+ ALL but lacks its diagnostic hallmark (Ph chromosome and BCR-ABL1 gene rearrangement) and carries instead other genetic aberrancies (frequently CLRF2+ and abnormal tyrosine kinase and JAK/STAT pathway activation). Of note, independent European and North American studies in large patient series disclosed an incidence of 25–30% for Ph-like ALL in the 20–39 years range, which was higher than that observed in younger and older patients, respectively [22].

In addition, copy number alterations (CNA), i.e., gene deletion or inactivation affecting several molecular pathways regulating cell proliferation and apoptosis response, may occur and bear prognostic relevance. Single or multiple CNA were detected in association with major genetic abnormalities. This exerted a pejorative prognostic effect, particularly with IKZF1 (Ikaros) and CDKN2A/2B deletions and others [23], such as in IKZF1^plus^ BCP ALL, in which IKZF1 deletions co-occurred with CDKN2A/2B, PAX5 or PAR1 deletions conferring the worst outcome [24]. The analysis of CNA was recently incorporated in some of the most advanced prospective risk models (see Section 2.3).

Due to its relative rarity, the age distribution of genetic/cytogenetic abnormalities and their prognostic significance are much less well known in AYA T-ALL. By inference with data collected in T-ALL in general, a novel four-gene prognostic classifier may reflect poor risk genetics (unmutated NOTCH1/FBXW7 and/or RAS and PTEN abnormalities) [25,26], whereas for many other genetic abnormalities detectable in T-ALL no firm prognostic significance has been established. Approximately 15% of T-ALL cases display an early-thymic precursor (ETP) immunophenotype with absent/weak CD5 expression, cross-lineage expression of immature myeloid markers and a typical, dysregulated gene expression profile (JAK/STAT, FLT-3, BCL-2, etc.). ETP ALL is considered a potential high-risk ALL entity [27,28,29].

### 2.3. Treatment: Traditional Adult vs. Modern Pediatric Regimens

Although it is not the main focus of the present review, the importance of an optimal treatment regimen for AYA ALL must be correctly understood. Some excellent reviews have recently been dedicated to this topic [6,16,17]. A summary of the evidence collected in comparative and non-comparative trials assessing the feasibility and efficacy of “pediatric-type” chemotherapy in AYA and adult ALL is presented in Table 2. The trials herein considered were selected on the basis of patient number (minimum of 50) and timing for data analysis and outcome reporting (minimum of 3 years) and are presented according to increasing patient age groups from younger AYA only to adult trials including AYA patients.

Overall, the available information generated by the use of “pediatric” treatments in AYA ALL can be summarized as follows:Intensive chemotherapy regimens inspired to modern pediatric schedules and treatment principles are superior to historical adult-type programs, as demonstrated with very few exceptions by comparative analyses among successive Phase 2 trials and many large non-comparative trials [6,16,17]. Whenever available, retrospective comparisons with historical datasets (not shown in the table and available in study references) confirm an average improvement of outcome measures of 15–25%.In these modern AYA or AYA-containing adult ALL studies, the projected survival rates at 5 years (range 3–7 years), assuming “cure” for most patients who remain disease-free at ≥5 years, is 50% and greater (overall survival, OS), with age-related variations and OS rates of 60–70% and occasionally higher in younger age groups.Unlike OS, which reflects the cumulative survival effect of both first line and salvage therapies, relapse-free and event-free survival (RFS and EFS) depict the curative potential of upfront therapy only, in CR patients and all study patients, respectively. These figures range 55–70% (RFS) and 40–74% (EFS), once again with significantly better results in younger age groups.The overall chemotherapy intensity is increased in pediatric-based regimens, with regard to vincristine, corticosteroids, antimetabolites (cytarabine, methotrexate and 6-mercaptopurine), L-asparaginase and, more recently, Pegylated-asparaginase (Peg-ASP). Consequently, drug-related toxicity may be higher, requiring higher clinical skills for the management and prevention of toxic side effects.The improved pediatric-like protocol may consist of an unmodified or modified pediatric schedule, in the latter case adapting some treatment elements to an increasing patient age with attending risks of treatment toxicity. The issue of Peg-ASP dosing and toxicity is highly critical in patients at older age [52,53].Contrary to younger patients, it appears difficult to demonstrate an advantage by pediatric-type regimens in patients older than 55 years [44,47,51]. Non-AYA patients become progressively less tolerant to intensive treatment and display a higher incidence and severity of toxic side effects.The patients who achieve CR, namely about 90% of all patients (≥95% in younger age groups), are usually risk-stratified to assess the individual risk class and decide about the application of risk-specific treatments that range, for high-risk (HR) patients, from chemotherapy intensification to allogeneic hematopoietic cell transplantation (HCT) and/or experimental new agents. Most patients at standard-risk (SR) or intermediate-risk can achieve cure with a full chemotherapy regimen including maintenance as standard of care, without HCT, this lowering the incidence of non-lethal and lethal toxicities (10–15% average mortality from HCT).In the risk stratification process, by analogy with pediatric trials, the analysis of post-induction MRD is crucial since it has been demonstrated to be the most powerful predictive factor for marrow relapse in multivariable analysis from several studies [7,8,9,10]. Therefore, MRD is currently used together with other risk factors for the definition of risk groups and individual risk profiles.

With regard to MRD-based risk stratification, the choice of MRD study method and time-point(s) as well as the layout of the final risk model have been quite variable across trials. In some studies, MRD was used alone or in combination with a minimal set of variables (i.e., very high-risk genetics such as KMT2A rearrangements), while in others it was part of complex risk models involving multiple risk factors. These differences are reported in a review article on the risk stratification criteria adopted by 11 European adult ALL Study Groups to orientate an allogeneic HCT decision (Table 3) [54]. This survey documents a consistent use of MRD (11/11, 100%) together with the frequent addition of selected adverse genetic/cytogenetic abnormalities (8/11, 73%) to support a diagnosis of HR ALL with an indication for allogeneic HCT.

The increasing precision and complexity of MRD-based risk stratifications is further illustrated in another risk model from an ongoing International pediatric and AYA project (age range 1–45 years), i.e., the ALLTogether study, in which the very large enrolment basis and thorough diagnostic work-up allow very accurately refining the patient risk class for risk-oriented treatments (Figure 1) [55].

Quite interestingly, in this study conceived by a childhood ALL Consortium, the diagnosis of T-ALL and a patient age >16 years qualified for an intermediate-high risk classification even with undetectable post-induction MRD and/or lack of other risk factors, to reflect the long-lasting notion of a worse outcome expected for patients aged ≥15 years in the pediatric setting. Unlike this, age or T-ALL diagnosis do not represent a clear risk classifier in most adult/AYA trials, although some differences are reported in favor of teens and AYA < 5 years (Table 3). The ALLTogether trial design highlights the place of a mixed MRD/genetic prospective risk classification that influences the chemotherapy intensity from SR to HR groups and guides the application of an allogeneic HCT or the randomized evaluation of additional risk/subset-specific therapeutic elements in intermediate/high-risk patients (CAR-T cells, tyrosine kinase inhibitors and immunotherapy), many of whom express high MRD levels, as well as treatment de-escalation in SR patients.

Combined risk stratification systems merging MRD and genetics were previously assessed in both childhood and adult ALL, enhancing the accuracy of risk stratification [23,26,56,57]. The UKALL-14 adult study introduced a novel prognostic index (PI_UKALL_) integrating the WBC count and patient age as continuous variables with cytogenetics/genetics and two post-induction MRD time-points. The PI_UKALL_ prognostic classification was a powerful predictor of outcome following chemotherapy and HCT in SR and HR patients, respectively [57]. The PI_UKALL_ was successfully tested in a very large retrospective COG series of more than 21,000 children and adolescents [58]. In the era of precision medicine, risk assessment is an evolving process that depends equally on the optimization of MRD analysis and its interaction with ALL biology. MRD detection can be performed by flow cytometry or by a molecular approach.

## 3. Methods of MRD Assessment

### 3.1. Multiparameter Flow Cytometry

Multiparameter flow cytometry (MFC) is based on a panel of monoclonal antibodies that bind specific cell surface markers useful for distinguishing normal cells from leukemic cells, identifying the leukemia-associated aberrant immunophenotype (LAIP). This method was standardized and constantly refined by the EuroFlow Working Group within the ESHLO Consortium (European Scientific Foundation of Laboratory Hemato-Oncology) which also includes the EuroClonality and EuroMRD European laboratory Groups working on different aspects of leukemia and lymphoma characterization and monitoring [59]. Flow cytometry is faster than the molecular approach but less sensitive (sensitivity of 10^−4^, i.e., the ability to detect one leukemic cell among 10,000 normal cells, or 0.01%). More recent studies using highly sophisticated eight-color MFC reported an increased sensitivity for MRD detection in B-precursor ALL, up to between 10^−5^ and 10^−6^ [60].

### 3.2. PCR for Fusion Genes and Transcripts

One third of ALL patients present chromosomal translocation-derived fusion genes, including BCR-ABL1, TCF-PBX1, KMT2A-AFF1 or KMT2A and other partner genes and ETV6-RUNX1 [61]. These chromosomal abnormalities are highly stable over time, therefore are good markers for MRD detection during and monitoring. The MRD quantification is obtained by qRT-PCR (quantitative Reverse Transcription Polymerase Chain Reaction) that compares fusion gene levels detected in a follow-up sample to a standard curve of plasmid containing chimeric-transcript at fixed concentrations [62].

### 3.3. PCR for Ig and TCR Gene Rearrangements

If the patients have been tested negative for chromosomal translocation at diagnosis, the molecular gold standard for MRD monitoring is the amplification of T-cell Receptor (TcR) and Immunoglobulin (Ig) gene rearrangements by Real Time quantitative PCR (RQ-PCR). Unlike the chimeric transcript, Ig/TcR are not directly involved in leukemia pathogenesis, but they mirror physiological events that occur during the ontogenesis of B and T lymphocytes. During lymphocyte differentiation, random insertions/deletions of nucleotides at the junctional sites of V (Variable)–D (Diversity)-J (Junctional) gene segments result in a diversity of antigen receptor; in the case of neoplastic evolution, all the tumor cells will express the same receptor sequence. Therefore, these insertions/deletions are specific to each patient and represent a fingerprint of leukemia. This approach, according to BIOMED-2 protocol [63], includes the identification at diagnosis of V–D-J regions of Ig and TcR gene rearrangements by PCR followed by heteroduplex analysis to distinguish between polyclonal and clonal, leukemia specific rearrangements. Nucleotide sequences of the identified clonal rearrangements are obtained by Sanger method. Allele Specific Oligonucleotides (ASO primer) (Figure 2) are then designed on these leukemia-specific sequences and used in combination with family-specific primers and fluorescent probes to identify and quantify leukemia cells by Quantitative PCR (ASO-qPCR). The fluorescence emission detected by the instrument in a sample is directly proportional to the amount of target DNA present.

Leukemia quantification in a follow-up sample is derived by comparing the measured fluorescence to serial dilutions of diagnostic material in a pool of mononuclear cells derived from eight healthy donors (standard curve). The MRD result is expressed as the logarithmic reduction compared to diagnosis. To ensure the monitoring of possible multiple leukemia clones, at least two ASO primers should be developed for each patient with a desirable sensitivity of 10^−4^ or 10^−5^ (i.e., the ability to detect one leukemic cell among 10,000 or 100,000 normal cells, which is 0.01% and 0.001%, respectively). This method has been developed and standardized by the EuroMRD/EuroClonality Consortium that, during the last 20 years, established precise rules to define both sensitivity of the assay and positivity or negativity of follow-up samples [64]. However, this method has some limitations including time consuming and complex procedures that only specialized laboratories can properly perform. It also requires a good quantity of diagnostic material to be used for standard curve in each follow-up evaluation. In addition, about 5–10% of patients do not have a leukemia specific probe either because no Ig/TCR rearrangements are detected at diagnosis or because the unique VDJ portion is very short or the designed ASO primer is not sufficiently specific and sensitive.

### 3.4. Next Generation Sequencing

Next-generation sequencing (NGS) is a new method of high-throughput DNA sequencing allowing to overcome some limits of the standard molecular approach. The EuroClonality-NGS working group developed an amplicon-based protocol for Ig/TcR marker identification in ALL. This assay employs a two-step PCR: in the first step, the most common Ig/TcR gene rearrangements are amplified in multiplex reactions with family specific primers (eight reactions), while, in the second step, short forward and reverse sequences (adaptors) are added to uniquely identify single patients and to allow the subsequent reaction phases. Then, a pool of amplification PCR products is created and sequenced by the Illumina or Ion Torrent platform [65,66,67]. An alternative, capture-based approach has also been described [68,69]. In this method gene rearrangements are not amplified by PCR but hybridized with probes that recognize all V, D and J of all known Ig/TCR regions, including IG Lambda and TcR Alpha that are usually difficult to amplify. This approach can also detect rarely used V/D/J portion for which no primers have been designed in PCR-based methods. Sequences generated by NGS approaches can be analyzed to identify leukemia specific markers by bioinformatics tools, including the ARResT/Interrogate web tool [70] (created by Euroclonality-NGS Consortium) and Vidijil (free tool) [71].

NGS-based clonal identification is routinely used in most MRD reference laboratories because it allows a faster clonal marker identification starting from a small quantity of diagnostic material. It also increases the ability to discriminate rearrangements difficult to resolve by Sanger sequencing. NGS has been reported to be successful also in MRD monitoring [72] and has been described to be at least as sensitive as standard RQ-PCR and to provide MRD quantification without the need of diagnostic material in each assay (standard curve). Furthermore, it allows the certain discrimination between residual leukemia cells and normal lymphocytes that can have similar but not identical rearrangements. However, this approach is still lacking standardization within collaborative groups to be properly applied in treatment protocols. The EuroMRD-NGS group is working on this aspect also introducing an internal quantification reference to allow comparability of results in different MRD laboratories. Furthermore, the available NGS assays for MRD are more expensive and the time to result is longer than ASO-Q-PCR when applied on a routine basis in reference laboratories working on a high number of samples within clinical trial.

### 3.5. Digital Droplet PCR

A promising, effective, low cost, new third-generation PCR for absolute MRD quantification is named digital droplet PCR (ddPCR) [73]. This technique can be applied in leukemia cases in which diagnostic material was sufficient to identify clonal rearrangements but not for preparation of standard curve for MRD quantification, thus limiting the possibility of patients monitoring over time. The analysis is performed by partitioning the follow-up sample amplification into many independent PCR, by inclusion of reagents (ASO fluorescent assay) and DNA into small droplets created by emulsion. The partitioning is a random event following the Poisson distribution. Therefore, with the production of high number of droplets, there is the probability to have either zero or a single molecule of target rearrangements. After amplification, there will be fluorescent droplets (i.e., with PCR product inside) or no fluorescence. The analysis software for the ddPCR will count the number of fluorescent events and will express the result as number of copies of template per microliter of reaction (copies/μL), taking into account the final volume of reaction. This will result in an absolute MRD value.

## 4. MRD in AYA Ph− ALL

### 4.1. MRD Study Results for Risk Stratification

MRD study results from representative clinical trials in AYA patients or AYA plus adults when the two groups were treated together are shown in Table 4. Depending on the response definition adopted by each study, MRD cut-off values separating SR from HR patients in a risk-oriented studies ranged from totally negative using highly sensitive molecular probes to <10^−3^ (<0.1%), whereas MRD detection timepoints were set from as early as Day 21–24/EOI (end of induction) to treatment Weeks 16–22. The MRD-based risk stratification was quite often unrelated to the initial risk profile (SR or HR) and motivated the choice of an allogeneic HCT in case of MRD persistence. In some of these trials, however, the MRD analysis was available for a limited proportion of patients, which is one half or less of all CR patients, and/or was not clearly or not always meant to guide a risk-oriented approach with HCT, left to the discretion of treating physicians or indicated for very high-risk conditions such as t (4;11)+ ALL, etc. Despite these discrepancies that may affect to some extent trial result interpretation and inter-trial comparability, MRD was universally recognized as a major determinant of outcome supporting risk-oriented treatment decisions.

**Table 4 cancers-13-02108-t004:** MRD_pos_-based results (risk stratification, risk-oriented therapy and clinical outcome) in AYA and adult Ph− ALL trials.

Trial/Study (Ref.)	Patient Age (Years), Median (Range)	MRD Analysis		Favorable MRD Response		Comparative Outcomes: Favorable MRD Cut-Offs Vs. Not
		Evaluable/CR, No. (%)	Method ^1^	Cut-Off ^2^	No. (%)	
AYA only (maximum age 40 years)						
MRC-UKALL2003 [11]	NR (16–24)	223/229 (97)	Mol	Negative/<0.01% d29, negative at EOC	54 (24)	5-y EFS 93% vs. 63–71% (*p* = 0.0001) ^3^
PETHEMA ALL08 [35]	20 (15–30)	61/68 (90)	MFC	<0.1% w5–6, <0.05% w19–20	48 (77)	NR
GIMEMA LAL1308 [39]	NR (18–35)	64/68 (94)/ 66/68 (97) 49/68 (72)/ 50/68 (73)	MFC Mol	<0.1% d33/78	37 (58)/54 (82) 28 (57)/38 (76)	4-y OS by d33 MRD 67–75% vs. 27–41% (*p* = 0.002) ^4^ 4-y RFS by d33 MRD 67–73% vs. 27–43% (*p* ≤ 0.025) ^4^ 4-y OS by d78 MRD 74–77% vs. 31–39% (*p* ≤ 0.01) ^4^ 4-y RFS by d78 MRD 71–72% vs. 26–34% (*p* ≤ 0.01) ^4^
MDACC (aBFM/HyperCVAD) * [36]	22 (13–39)	93/199 (47)	MFC	<0.01% d29/d84	58 (62)	5-y OS by d29 MRD 75% vs. 40% (*p* = 0.004) 5-y OS by d84 MRD 75% vs. 22% (*p* = 0.0004)
CALGB 10,403 [37]	24 (17–39)	80/237 (34)	Mol	Negative at EOI	35 (44)	3-y RFS 85% vs. 54% (*p* = 0.0006)
AYA and adults (maximum age > 40 years)						
NOPHO 2008 [14]	26 (18–45)	NR/218	Mol	<0.1% d29/d79	(56–64) ^5^	NR
GRAALL 2003–2005 [26]	31 (15–59)	423/860 (49)	Mol	Negative/<0.01% w6	265 (63)	5-y CIR 23–31% vs. 60% (*p* ≤ 0.01)
GMALL 07/03 [74]	34 (16–65)	1057/1857 (57)	Mol	Negative w16	625 (59)	5-y OS 83% vs. 43–68% (*p* < 0.0001) ^6^
MDACC [75]	37 (15–86)	215/394 (55)	MFC	<0.01% d24/d108	147 (68)/194 (90)	3-y OS by d24 MRD 76% vs. 49–61% (*p* = 0.001) ^7^ 3-y EFS by d24 MRD 65% vs.16–46% (*p* < 0.001) ^7^
PETHEMA ALL-HR11 [49]	40 (15–60)	286/289 (99)	MFC	<0.1% w5–6	220 (82)	5-y OS 59% vs. 38% (*p* < 0.001)
NILG 10/07 [51]	41 (18–65)	109/140 (78)	Mol	<0.01% w10–16/negative w22	68 (62)	5-y OS 78% vs. 34% (*p* < 0.0001) 5-y RFS 66% vs. 29% (*p* < 0.0001)
HOVON-100 ** [41]	42 (18–70)	168/297 (56)	Mol/MFC	<0.01% after consolidation 1	126 (75)	NR

Abbreviations: CALGB, Cancer and Leukemia Group B; GIMEMA, Gruppo Italiano Malattie Ematologiche dell’Adulto; GMALL, German Multicenter Study Group for Adult ALL; GRAALL, Group for Research on Adult Acute Lymphoblastic Leukemia; HOVON, Hemato-Oncology Foundation for Adults in the Netherlands; MDACC, M.D. Anderson Cancer Center; NILG, Northern Italy Leukemia Study Group; NOPHO, Nordic Society of Pediatric Hematology and Oncology; PETHEMA, Programa Español de Tratamientos en Hematología; RALL, Russian ALL Study Group; MRC-UKALL, Medical Research Council-United Kingdom ALL Study Group; EFS, event-free survival; OS, overall survival; RFS, relapse-free survival; CIR, cumulative incidence of relapse; NR, not reported (or not available); ^1^ Mol, molecular assays; MFC, multiparametric flow cytometry; ^2^ d, day; w, week; EOC, end of consolidation; EOI, end of induction; ^3^ observed ranges in MRD positive (>0.01% d29) or indeterminate (<0.01% at EOC); ^4^ observed ranges according to MRD timepoint by MFC or Mol method; ^5^ d29 MRD response rates according to ALL subset (BCP ALL < 100 WBC, B-ALL > 100 WBC, T-ALL); ^6^ observed range > vs. ≤ 10^−4^; ^7^ observed ranges according to d108 MRD status; * cumulative MRD-based results from both treatment regimens; ** update on all study patients (personal communication by Dr. A.J. Rjineveld, Rotterdam, The Netherlands).

### 4.2. Terminology of MRD Response for Clinical Purposes

According to technical terminology, an MRD “negative” status (MRD_neg_) should be intended as an undetectable/unmeasurable MRD using highly sensitive molecular probes (sensitivity 10^−4^ to 10^−5^) or comparable or only slightly less sensitive MFC techniques. In the clinical setting, however, a broader definition of favorable MRD response has been applied until now, from a molecular MRD < 0.1%/<10^−3^ evaluated at an early time-point to a combination of different MRD reads at different time-points, namely < 0.1–0.01%/< 10^−3/−4^ early on to <0.01%/< 10^−4^ and MRD_neg_ afterwards. Consequently, uniform MRD terminology for clinical use is presently lacking for adult/AYA ALL trials, although suggestions were provided by expert panels for an MRD ≥ 0.01%/≥ 10^−4^ at about 12 treatment weeks (i.e., following at least three intensive chemotherapy courses) to represent true high MRD associated with high risk of recurrence (MRD_pos_). Moreover, within the MRD_pos_ group, increasing MRD levels from 10^−4^ to 10^−1^ are predictive of an increasingly worse survival [76]. On the contrary, an MRD < 0.01%/< 10^−4^ has been consistently associated with the best clinical outcome independently of treatment type and ALL subset, with minor prognostic differences among MRD_neg_ and MRD < 0.01%/10^−4^ patient groups. The 0.01%/10^−4^ threshold may therefore be considered the current benchmark for operational definitions in clinical trials, while the distinction of a MRD_neg_ status should be maintained to identify the best MRD response and prognostic subset.

### 4.3. MRD-Related Outcomes

As shown in Table 4, whichever the risk stratification, the risk-oriented approach and the MRD study method, the patients who displayed persistent MRD following CR induction and/or early consolidation chemotherapy fared significantly worse than MRD responders.

However, with few exceptions, MRD was not assessable in about one fourth of CR patients, due to either an insufficient diagnostic and/or follow-up sampling or a failure to generate a molecular probe, while, in this regard, the search for a case-specific LAIP was less troublesome, with success rates >95% when it was applied systematically [35,40,49]. Performing an adequate ALL cell sampling may become a highly critical issues, because, in some studies, only 50% or less of the patients underwent MRD analysis [26,37], or this same figure was slightly above 50% in others [41,44]. These low rates of MRD analysis were more frequent in adult trials and using molecular MRD assays. In the end, in MRD evaluable patients, the average rate of a favorable MRD response after induction-consolidation was about 60–70%, with a trend to higher figures in AYAs. About 35–50% of all patients achieved a significant MRD reduction early on (collectively at end of induction, EOI) and fared better than others in many reports, mirroring the comparable, much larger experience in childhood ALL.

With regard to MRD-associated outcomes, the data in the table document the substantial prognostic advantage associate with a negative or favorable MRD course as defined in each study. Overall survival probabilities for MRD responders were in the range of 60–70% and greater, with variations by age group and other, including an early or late MRD response. A pertinent example concerning EOI MRD is illustrated by a sub-analysis of the NILG 10/07 trial [50] in 61 AYA patients aged 18–40 years. While the relapse risk was 24% at 5 years in the 42 MRD_neg_ patients (69%), those with an EOI/Week 4 MRD < 0.01%/10^−4^ confirmed at Week 10 experienced the lowest relapse incidence (15%) with an excellent CR duration (85% at 5 years) (Figure 3).

Because treatment failure was most commonly caused by an ALL relapse, which in turn is strongly predicted by MRD_pos_, two main issues deserve to be further elucidated: an “unexpected” recurrence in MRD_neg_ patients and the exact role of an allogeneic HCT salvage (and other therapies) in MRD_pos_ patients.

### 4.4. Risk of Relapse in MRD Responders

Virtually all MRD-based trials and studies reported a fraction of treatment failures in MRD responsive patients, from 10% to 30–35%, depending on associated risk factors (patient age, genetics, ALL subset). This MRD_neg_ group at increased risk of relapse constitutes a diagnostic and therapeutic challenge. The more likely explanation for this occurrence could lie in a combination of technical issues and disease biology. Because the sensitivity of the best available techniques currently available for large-scale clinical application does not exceed 0.001%/10^−5^, some MRD_neg_ patients could still harbor undetectable residual ALL cells able to induce a subsequent relapse. In addition, genetically adverse ALL subsets do worse even if MRD_neg_, as already demonstrated by some studies [23,26,56,57]. This suggests that persistence of unmeasurable MRD may be more frequent in some ALL subsets than others, leading to practical considerations for the design of improved risk-oriented strategies, as discussed below.

### 4.5. Allogeneic HCT for MRD Positive States

Thus far, an allogeneic HCT has been the preferred therapeutic option for MRD_pos_ patients in risk-oriented trials, to avert the high risk of recurrence expected with the use of chemotherapy only. This indication is shared by the American Society for Transplantation and Cellular Therapy and by all European Study Groups including AYA patients, as stated in an EBMT position paper [54,77]. Another option made available to younger AYA patients in some trials has been further chemotherapy intensification, however with final results below those obtained in MRD_neg_ patients and/or difficult to interpret due to small patient groups. Currently, the MRD status should to be integrated with ALL genetics and other HR clinical characteristics (such as in Ph-like ALL and ETP-ALL) in the decision-making model driving patients to allogeneic HCT. Consequently, the role of HCT will have to be reassessed combined these new risk definitions, as well as pre-transplantation therapy with novel chemo-immunotherapy combinatory regimens aiming to induce a MRD_neg_ pre-HCT condition (Section 5).

#### 4.5.1. Allogeneic HCT Results in MRD_pos_ AYA Ph− ALL

HCT results obtained in HCT-eligible AYA patients identified through an MRD-based risk stratification and/or other HR criteria are reported in Table 5. Most of these trials reported a significant improvement of EFS/OS with HCT compared to no HCT in these patients, in both AYA and non-AYA groups. Moreover, these trials were not specifically designed to compare allogeneic HCT to other therapeutic interventions in MRD_pos_ patients. Several other retrospective studies and meta-analyses, quite heterogeneous as far as HR definition, population age and MRD evaluation, highlighted the advantage of HCT over intensive-pediatric based chemotherapy [8,78,79,80,81,82], even if HCT results were quite variable and to some extent suboptimal because of the occurrence of HCT-related death and post-transplantation relapse in MRD_pos_ patients (long-term OS: 45–70% with vs. ≤25% without).

**Table 5 cancers-13-02108-t005:** Application and results of allogeneic HCT or other intensification protocols for MRD positive and HR states in AYA/adult Ph− ALL.

Trial/Study (Ref.)	MRD+ and/or HR Patients Eligible to HCT/HR Protocol (No.)	Had Allogeneic HCT/HR Protocol (No.)	Outcomes *
AYA only (maximum age 40 years)			
PETHEMA ALL08 [35]	2 MRD_pos_ and 20 HR to HCT or HR protocol	5 HCT and 13 HR protocol	4 HCT survivors (80%) and 7 HR protocol survivors (54%)
MRC-UKALL2003 [11]	109 MRD_pos_ to random study and 14 HR to HCT ^1^	64 randomized and 14 HCT	9 HCT survivors (64%)
MDACC [76]	17 MRD_pos_ or HR to HCT	17 HCT	7 survivors (41%), 5 in CR (29%)
CALGB 10,403 [37]	20 HR/other ^2^ to HCT	20 HCT	8 survivors (40%)
GIMEMA LAL1308 [40]	21 MRD_pos_ to HCT and 9 HR to HR protocol	15 HCT and 12 HR protocol	4-y OS HR 52.6% vs. SR 73.4% (*p* = 0.032) 4-y RFS HR 54.2% vs. SR 66.6% (*p* = 0.51)
AYA and adults (maximum age > 40 years)			
NOPHO 2008 [14]	35 MRD_pos_ to HCT and 45 HR to HR protocol	NR	5-y EFS 61% ^3^
GRAALL 2003–2005 [83]	105 HR MRD_pos_ to HCT	59 HCT	3-y OS 65% vs. 40% (*p* = 0.001) 3-y RFS 56% vs. 22% (*p* = 0.002)
GMALL 07/03 [74]	196 MRD_pos_ to HCT	121 HCT	5-y OS 53% vs. 28% (*p* < 0.0001) 5-y CRD 56% vs. 9% (*p* < 0.0001)
PETHEMA ALL-HR11 [49]	66 MRD_pos_ and 40 HR to HCT	62 HCT	5-y OS 54% (as treated)
NILG 10/07 [51]	41 MRD_pos_ to HCT	23 HCT	5-y OS 35% vs. 14% (*p* = 0.02) 5-y OS RFS 43% vs. 12% (*p* = 0.09)

Abbreviations: NR, not reported; CRD, CR duration; HR, high-risk; SR, standard risk. * HCT vs. no HCT patients (when available, if not differently indicated); ^1^ random study: standard vs. intensified chemotherapy (Vora A et al., Lancet Oncol 2014;15: 809–818); unknown MRD status of HR HCT patients; ^2^ at physician’s discretion; ^3^ with/without HCT.

#### 4.5.2. Pre-Transplantation MRD Status

In MRD_pos_ patients, a key time-point for MRD evaluation is just before an allogeneic HCT. MRD positivity at transplantation is the most powerful predictor of relapse and poor outcome, as evidenced by a meta-analysis [84] evaluating 21 retrospective or prospective studies published 1998–2016, all including AYA patients. The pooled results evidenced a higher relapse risk (hazard ratio 3.26; *p* < 0.05) and lower RFS (hazard ratio 2.53; *p* < 0.05) in patients with positive pre-transplant MRD in comparison to those with negative MRD. Further studies published 2018–2020 in different transplant settings (related and unrelated donor or haploidentical HCT) and mixed age populations with large percentages of AYA, confirmed the unfavorable prognostic role of pre-HCT MRD_pos_, with a relapse incidence of 32–73% vs. 19.7–24% in pre-HCT MRD_neg_ subsets [85,86,87,88,89,90,91]. Thus, achieving an MRD_neg_ status before HCT in MRD_pos_ patients should improve the overall transplantation outcome. The role of new targeted immunotherapy in this setting is discussed in the following section.

## 5. New Therapeutic Options for MRD Positive ALL

### 5.1. Immunotherapy for BCP ALL

#### 5.1.1. Blinatumomab

Blinatumomab, the first bispecific T-cell engager (BiTE), has been approved for relapsed/refractory (R/R) ALL. In this setting, blinatumomab proved more effective and better tolerated than conventional chemotherapy and able to induce molecular remissions [92]. For its favorable safety profile and mechanism of action, blinatumomab represents the ideal treatment of MRD, and, in fact, it has been extensively tested in this setting both in first or later CR. In two subsequent Phase II clinical studies conducted in MRD_pos_ adult ALL patients in hematologic CR but with a high MRD level (≥10^−3^), a single cycle of blinatumomab induced a major MRD response in about 80% of the patients [93,94]. Based on these results, the United States Food and Drug Administration (FDA) and European Medicines Agency (EMA) both approved blinatumomab as the first drug registered for the treatment of MRD.

Patients treated for MRD positivity in first CR and achieving MRD negativity most frequently are referred to HCT as further consolidation. However, for the time being, there is no evidence that OS is better in patients who did or did not undergo transplantation. On the contrary, the outcome of patients receiving blinatumomab in second or later CR and did not proceed to HCT, proved to be inferior [94].

A real-world effectiveness and safety study of blinatumomab in R/R Ph− B-ALL patients has been recently conducted in Europe. In total, 118 patients were included with a median age of 45 years; 22% had previous HCT. Within two blinatumomab cycles, 74% of patients achieved CR or CR with incomplete/partial hematologic recovery: among 44 evaluable patients, 45.5% had a complete MRD response. The majority (78%) of responders proceeded to alloHCT. The estimates for RFS and OS at 24 months were 50% and 58%, respectively [95].

The impact of blinatumomab on MRD has been recently addressed in high-risk first-relapse childhood BCP ALL. In this study, patients were randomized to receive one cycle of blinatumomab (15 μg/m^2^/d for four weeks, continuous intravenous infusion) or chemotherapy as third consolidation before allogeneic HCT. After a median of 22.4 months of follow-up, the incidence of events in the blinatumomab vs. consolidation chemotherapy groups was 31% vs. 57% (log-rank *p* < 0.001). MRD remission by PCR was observed in 90% of patients in the blinatumomab group and in 54% in the consolidation chemotherapy group [93]. In a second randomized study, the effect of post-reinduction therapy consolidation with blinatumomab vs. chemotherapy was evaluated on DFS in children and AYAs with first relapse of B ALL. The 2-year DFS rate was 54% for the blinatumomab group vs. 39% for the chemotherapy group. This difference was considered not statistically significant (one-sided *p* = 0.03). The 2-year OS rate was 71% for the blinatumomab group vs. 58% for the chemotherapy group. This difference was statistically significant (one-sided *p* = 0.02). After the first cycle of randomized therapy, the MRD_neg_ rate was 75% for the blinatumomab group vs. 32% for the chemotherapy group (*p* < 0.001). Finally, for the blinatumomab group, 70% proceeded to allogeneic HCT, compared with 43% for the chemotherapy group (*p* < 0.001) [96].

#### 5.1.2. Inotuzumab Ozogamicin

Inotuzumab ozogamicin (InO), a humanized anti-CD22 monoclonal antibody conjugated to the cytotoxic antibiotic calicheamicin has shown strong single agent activity in R/R BCP ALL patients [97]. In this study, patients who received InO versus standard chemotherapy achieved greater remission and MRD_neg_ rates as well as an improved OS. Compared with MRD_pos_, MRD_neg_ status with CR or CR with incomplete hematologic recovery was associated with significantly improved OS and RFS, respectively. Median OS was 14.1 versus 7.2 months, in the MRD_neg_ versus MRD_pos_ groups [98]. InO is being studied as frontline consolidation drug in AYA Ph− ALL in a North American Phase 3 trial (NCT03150693, C10403 chemotherapy backbone +/− InO) and for MRD_pos_ IR patients in the new European ALLTogether children and AYA project [55].

### 5.2. Chimeric Antigen Receptor-Modified T-Cell Therapy

In clinical trials for R/R B-ALL, chimeric antigen receptor-modified T-cells (CAR-T) targeting CD19 (Tisagenlecleucel, CTL019) produced CR rates exceeding 80% to 90% and became the first CAR T-cell therapy approved by FDA in August 2017. The single-arm, multicenter, global registration trial (ELIANA) conducted across 25 centers demonstrated a CR rate of 81% in 75 patients with R/R B-ALL treated with tisagenlecleucel, with undetectable MRD achieved in 100% of responders. At 12 months, RFS and OS were 59% and 76%, respectively. Additional clinical trials have been conducted with other CAR-T cell products. While the rate of CR has been largely confirmed [99,100,101,102,103,104] in adult patients, the duration of response has been significantly less impressive.

A significant clinical benefit was observed particularly in responding patients who also achieved MRD_neg_ status. A subsequent allogeneic HCT has also been proposed as an effective strategy to avoid an early relapse after CAR-T cell therapy [105].

Based on these results, it is likely that even for CAR-T cells the greatest benefit could come from an earlier use in the setting of MRD. To validate this hypothesis, in 2019, the COG cooperative group decided to launch the AALL1721/Cassiopeia study, a Phase 2 single-arm trial of tisagenlecleucel in children and young adults in CR1 with high-risk criteria and persistent MRD (by flow cytometry) at the end of chemotherapy. The primary end point of this study is 5-year DFS.

### 5.3. Investigational Agents for T-ALL

T-ALL represents about 20% of ALL cases and compared to B-precursor ALL offers fewer opportunities for the exploitation of MRD-targeting new agents, partly because they cannot discriminate between normal or regenerating T-cells and residual T-lymphoblasts and could therefore cause an extreme T-cell suppression (fratricide) leading to lethal infections. At present, the clinical experience with immunotherapy and other targeting agents against T-ALL is limited and confined to relapsed/refractory disease, which is in T-ALL an extremely difficult therapeutic setting after the failure of current highly intensive first-line treatments. The only exception to that is nelarabine, which after proving relatively effective at salvage has been evaluated in untreated T-ALL, either frontline or following detection of MRD.

#### 5.3.1. Nelarabine for MRD_pos_ T-ALL

In a large randomized COG trial [106], the use of nelarabine improved the 4-year RFS vs. the no nelarabine arm (88% vs. 82%; *p* = 0.029). In this 4-arm study conducted in children and AYA 1–30 years, the best outcome was obtained in patients simultaneously randomized to nelarabine and Capizzi-style methotrexate (vs. no nelarabine and/or high-dose methotrexate), with a projected RFS of 91%. While other variably successful nelarabine-based trials have been performed [107] or are near to completion in AYA/adult T-ALL [17,108], a German study was specifically focused on MRD_pos_ T-ALL. In this experience, 6 out of 12 MRD_pos_ T-ALL patients achieved MRD negativity (50%) following nelarabine, some were transferred to HCT and only two relapsed [74]. All these data prompts further evaluation of nelarabine in MRD_pos_ AYA T-ALL, particularly in adverse subsets such as ETP-ALL, which is associated with a higher risk of MRD persistence [29,109]

#### 5.3.2. Immunotherapy for MRD_pos_ T-ALL

Some new immunotherapeutics could be profitably used against MRD in AYA T-ALL. The most promising agent for large-scale application is the monoclonal antibody daratumumab which is directed against the CD38 antigen, largely represented in T-lymphoblasts. Trials with daratumumab are ongoing, also in combination with chemotherapy, and will be informative about the induction of MRD negativity in clinically responsive patients [108]. In this regard, daratumumab effectively eradicated MRD in some small T-ALL series [110,111,112]. In a preclinical model, an anti-IL-7Rα monoclonal antibody was confirmed effective [113].

By analogy with B-lineage ALL, CAR-T cell therapy is also under investigation in T-ALL, although at a far earlier stage of development [17,108]. These first studies were mainly performed in patients with resistant or relapsing disease and involved anti-CD5, -CD7 and -T-cell receptor β CAR-T constructs. A new anti-CD7 CAR-T product generated by CRISPR/Cas9 gene editing technology demonstrated resistant to fratricide and was devoid of any alloreactive/graft-versus-host potential (UCART7) [114]. A comparable CAR-T product from China (TruUCAR GC027) induced four MRD remissions out of five AYA patients aged 19–38 years [115].

### 5.4. Other Experimental Approaches

#### 5.4.1. Molecular Profiling for Precision Medicine

A massive amount of data regarding ALL biology and the development of resistance to standard anti-ALL therapy has been generated in recent years [116,117,118,119,120,121]. These studies paved the way to experimental therapeutic attempts targeting altered molecular pathways [122,123,124,125]. Although the clinical experience is thus far limited and involves mainly the late stages of disease, this is a rapidly expanding area with some notable examples that may be considered for the management of MRD_pos_ states in AYAs.
Most relevant is Ph-like ALL, which is rather frequent in AYA patients and carries a higher risk of MRD persistence. Some of the associated gene abnormalities recognizable in this poor risk entity (*ABL*-class fusions, *CLRF2* deregulation and JAK/STAT and IL7R pathway alterations, among others) are actionable by TK inhibitors, JAK inhibitors (ruxolitinib) and other similar drugs. Trials in children, AYA and adults have been incepted worldwide, sometimes with promising preliminary results [126,127,128,129]. The data from these studies may elucidate which of these new drugs or drug combinations with either chemotherapy, immunotherapy and/or other targeting agents could optimize the outcome of the distinct genetically defined Ph-like ALL subsets.Among the BCP ALL subsets to consider for targeted therapy is t (v;11) + ALL or ALL carrying KMT2A gene rearrangements, most frequently t (4;11) + ALL. This entity stands out for its clinical aggressiveness. In this subset, the available molecular studies point to a therapeutic use of BCL-2 inhibitors (venetoclax and navitoclax) and DOT-L1 and histone-deacetylase inhibitors; however, the relative rarity of this ALL syndrome precludes an extensive clinical evaluation of these drugs outside large collaborative clinical trials.There are many more candidates for targeted therapy of BCP ALL subsets or ALL in general, as extensively reviewed [122]. Most of these drugs are under investigation in early clinical trials, and it is too early to define exactly their place and/or anticipate their approval for use as standard agents for front-line therapy and/or MRD_pos_ conditions. Worthy of mentioning are the proteasome inhibitors, again the BCL-2 inhibitors and the activators of P53-mediated apoptosis, given the frequent dysregulation of these molecular mechanisms. Likewise, the analysis of bone marrow immune cell contexture led to identify a poor risk ALL subset with PD1 + TIM3 + CD4 + bone marrow T-cells > 0.1% that might be targeted by PD1 checkpoint inhibitors [130].Many of the new drugs potentially active in BCP ALL could be exploited in T-ALL as well, namely inhibitors of the antiapoptotic BCL-2 family members and inhibitors of JAK/STAT, PI3K/Akt/mTOR, MAPK and Notch-1, the latter being a typical T-ALL target. While experience with Notch-1 inhibitors has been rather disappointing so far, BCL-2 inhibitors navitoclax and venetoclax induced a CR in six of 16 patients with refractory T-ALL, achieving undetectable MRD in four [129]. ETP ALL may be sensitive to the JAK-2 inhibitor ruxolitinib.

#### 5.4.2. Drug Sensitivity Profiling for Precision Medicine

An effective, functional drug screening is now obtainable through ex vivo models that employ extensive drug libraries detecting expected (based on prior molecular screening) or unexpected drug vulnerabilities, together with the evaluation of drug activity in patient-derived xenografts (PDX) (reviewed in [122]). These innovative studies may reveal and/or confirm highly promising single-agent or combinatory approaches with new targeting agents in specific ALL entities and/or individual patients. Effective drug combinations were identified for high-risk BCP ALL including Ph-like ALL (BCL-2 and MCL-1 inhibitors; TK inhibitors dasatinib and ponatinib) [131,132], T-ALL and ETP-ALL (ruxolitinib and dexamethasone; dasatinib; and venetoclax and bortezomib) [133,134]. This represents an exciting new area for treatment optimization of MRD_pos_ states.

## 6. Conclusions

The recent international experience in AYA Ph− ALL confirms that approximately 65% or more of these patients may achieve cure. This represents an outstanding therapeutic achievement, not very far from the 85–90% cure rate documented in children and highly encouraging given the different prognostic patterns and the increasing treatment complexity of AYA compared to childhood ALL.

In this field, MRD has emerged as a strong, dominant risk factor, necessary information by which we can modulate treatment intensity up to the level of allogeneic HCT or alternatively choose new treatment modalities (novel immunotherapeutics and new experimental agents) and design innovative risk- and MRD-oriented trials to improve even further the outcome of specific ALL entities.

More than 40 years ago, David Pinkel, a pioneer of ALL therapy, stated that “historically, when therapists have found themselves stymied in improving the prognosis of a disorder, new understanding of its biology has provided the key to further progress” [135]. Today, the study of MRD, which can be regarded as a behavioral marker of ALL biology with function of therapeutic target across all disease and patient subsets, continues to offer new chances of improving the outcome of AYA patients with ALL.

## Figures and Tables

**Figure 1 cancers-13-02108-f001:**
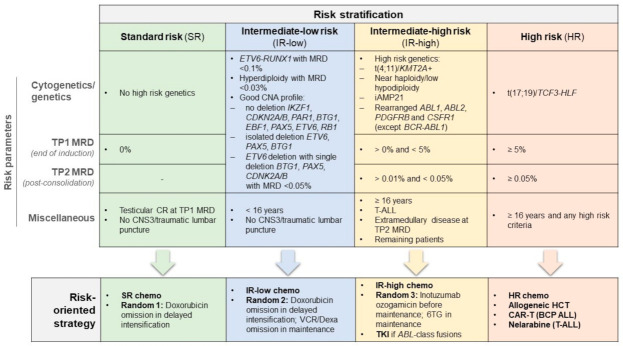
The ALLTogether MRD-based, mixed risk stratification system for risk-oriented therapy in patients with Ph− ALL 1–45 years. The master trial consisted of chemotherapy (chemo) of increasing intensity for SR, IR and HR patients, respectively, along with other risk-specific randomized or non-randomized interventions as indicated. TKI denotes tyrosine kinase inhibitor (imatinib) for cases with ABL-class fusions and CAR-T and BCP denote chimeric antigen receptor T-cell therapy and B-cell precursor ALL, respectively.

**Figure 2 cancers-13-02108-f002:**
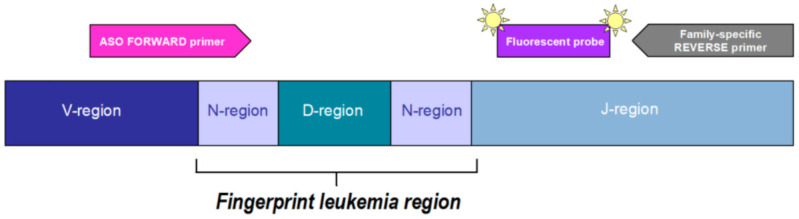
Leukemia specific assay design. An allele-specific oligonucleotide (ASO) primer is designed on the N-region between the V/D or D/J gene junction and used in combination with a family-specific primer and fluorescent probe. The fluorescence emission detected by the instrument is directly proportional to the amount of target DNA in the sample.

**Figure 3 cancers-13-02108-f003:**
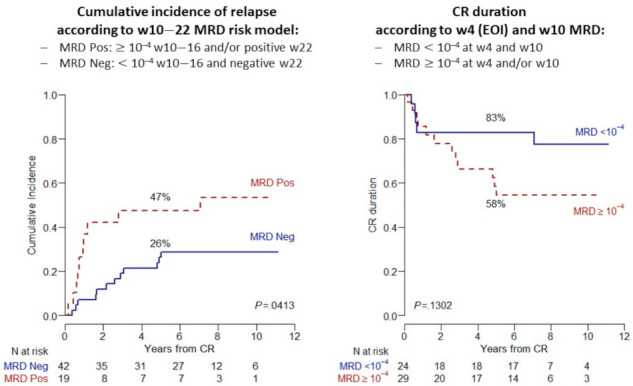
MRD-related incidence of relapse and duration of CR from NILG trial 10/07 in 61 AYA patients aged 10–40 years with Ph− ALL: (**Left**) cumulative incidence of relapse; and (**Right**) comparative CR duration in subsets with MRD < 0.01%/10^−4^ at EOI/Week 4 and Week 10 vs. others (non-significant *p* value due to small patient number).

**Table 1 cancers-13-02108-t001:** Summary table comparing the age-related incidence (percent of all evaluable cases) of main genetic/cytogenetic abnormalities in BCP ALL (data adapted from Roberts KG et al. [18]).

ALL Subset	Prognostic Category	Genetic/Cytogenetic Abnormality	Children (<15 Years)	AYA (15–40 Years)	Older Adults (>40 Years)
Ph−	Favorable	High hyperdiploidy	20–25%	5%	<5%
		t(12;21)/ETV6-RUNX1	25%	<5%	1%
	Intermediate	Normal Karyotype	10%	-	-
		t(1;19)/TCF3-PBX1	5%	<5%	1%
	Unfavorable	Low hypodiploidy,	1%	5%	>10%
		t(v;11)/KMT2A+	6%	4%	15%
		Ph−like	10–15%	25–30%	20%
Ph+	Unfavorable *	t(9;22)/BCR-ABL1	2%	6%	25%

* Turning standard/intermediate-risk with the introduction of TKI-based therapies.

**Table 2 cancers-13-02108-t002:** Key findings from selected trials of pediatric-based regimens for AYA and adult Ph− ALL. The projected overall survival (OS), relapse-free survival (RFS) and event-free survival(EFS) at ≥3 years are shown. Data are ordered according to increasing patient age groups.

Age Groups and Trials (Ref.)	Patient Age (Years), Median (Range)	No. of Patients	Outcome Estimates (y, Years)	OS (%)	RFS (%)	EFS (%)
Maximum age ≤ 25 years						
FRALLE-93 ^1^ [30]	15.9 (15–20)	77	5-y	78	72	67
CCG 1882/191 ^1^ [31]	16 (16–20)	197	7-y	67	-	63
JALSG ALL202-U [32]	19 (15–24)	139	5-y	73	67	-
CCG 19,61 ^1^ [33]	(16–21)	262	5-y	78	-	72
MRC UKALL 2003 [11]	(16–24)	229	5-y	76	-	72
Maximum age ≤ 40 years						
PETHEMA ALL-96 [34]	20 (15–30)	81	6-y	69	-	61
PETHEMA ALLRE08 [35]	20 (15–30)	66	5-y	74	-	-
MDACC (augmented BFM) * [36]	22 (13–39)	106	5-y	60	-	-
CALGB 10,403 [37]	24 (17–39)	295	3-y	73	66	59
FRALLE 2000-BT [38]	(15–29)	89	5-y	66	-	61
GMALL 07/03 [39]	(15–35)	887	5-y	65	61	-
GIMEMA LAL1308 [40]	(18–35)	76	4-y	60	60	-
HOVON-100 ^1^ [41]	(18–40)	159	5-y	60–56 ^2^	58	61–64 ^2^
Maximum age ≤ 55 years						
NOPHO ALL2008 [14]	26 (18–45)	221	5-y	78	-	-
DFCI 01–175 ^1^ [42]	28 (18–50)	92	4-y	67	69	69
DFCI 06–254 ^1^ [43]	32 (18–50)	89	3-y	75	73	73
GMALL 07/03 [44]	35 (15–55)	1226	3-y	60–67 ^3^	-	-
Maximum age > 55 years						
RALL 2009 [45]	30 (15–60)	250	4-y	66	69	-
GRAALL-2003 [46]	31 (15–60)	225	3.5-y	60	59	55
GRAALL-2005 [47]	36 (18–59)	787	5-y	59	-	52
Toronto (DFCI 91–01) [48]	37 (18–60)	85	5-y	63	71	-
PETHEMA ALL-HR-11 [49]	40 (15–60)	348	5-y	49	-	40
JALSG ALL 202-O [50]	40 (24–65)	115	5-y	64	58	-
NILG 10/07 ^4^ [51]	41 (18–65)	161	5-y	52	53	46

* Outcome not improved in comparison with historical, traditional adult-type treatment; outcome was comparatively improved in all other studies without an asterisk (figures not reported, available in references); ^1^ including a proportion of patients with Ph+ ALL (exact figures in study in trial reports); ^2^ by randomization arm (+/–Clofarabine); ^3^ referred to two different Pegylated-Asparaginase treatment cohorts; ^4^ outcome estimates for 135 patients 18–55 years: 5-year OS 60%, RFS 56%, EFS 52%. Abbreviations: CALGB, Cancer and Leukemia Group B; CCG, Children’s Cancer Group; DFCI, Dana Farber Cancer Institute; GIMEMA, Gruppo Italiano Malattie Ematologiche dell’Adulto; GMALL, German Multicenter Study Group for Adult ALL; GRAALL, Group for Research on Adult ALL; HOVON, Hemato-Oncology Foundation for Adults in the Netherlands; JALSG, Japan Adult Leukemia Study Group; MDACC, M.D. Anderson Cancer Center; NILG, Northern Italy Leukemia Study Group; NOPHO, Nordic Society of Pediatric Hematology and Oncology; PETHEMA, Programa Español de Tratamientos en Hematología; RALL, Russian ALL Study Group; UKALL, United Kingdom ALL Study Group.

**Table 3 cancers-13-02108-t003:** Summary of risk factors considered by 11 European adult ALL study groups for risk stratification orientating the choice of an allogeneic HCT in first CHR in Ph− ALL (all studies including AYA patients), adapted from [54]. For MRD levels defining MRD POS status.

Study Group	MRD	Genetics	WBC	Miscellaneous
RALL	+	KMT2A+, t(1;19)	-	Age > 30
GMALL	+	KMT2A+	>30 (B)	Late CR, proB, early/mature-T
HOVON	+	adverse	>30 (B), >100 (T)	Late CR
PALG	+	KMT2A+	>30 (B), >100 (T)	CNS+
FALL	+	Abn11q, hypodiploid	>100	Late CR, D15 BM blasts > 25%
GIMEMA	+	adverse	>100	Early/mature-T
UKALL	+	adverse	High counts	-
SVALL	+	KMT2A+, hypodiploidy	-	EOI BM blasts > 5%
CELL	+	-	-	-
PETHEMA	+	-	-	-
GRAALL	+	-	-	-

Complete list of adverse genetics/cytogenetics available in original study references. Abbreviations: WBC, white blood cells (×10^9^/L); BM, bone marrow; CNS, central nervous system; D, day; EOI, end of induction; CELL, Czech Leukemia Study Group; FALL, Finnish ALL Study Group; GIMEMA, Gruppo Italiano Malattie Ematologiche dell’Adulto; GMALL, German Multicenter Study Group for Adult ALL; GRAALL, Group for Research on Adult ALL; HOVON, Hemato-Oncology Foundation for Adults in the Netherlands; PALG, Polish Adult Leukemia Group; PETHEMA, Programa Español de Tratamientos en Hematología; RALL, Russian ALL Study Group; SVALL, Swedish Adult ALL Study Group; UKALL, United Kingdom ALL Study Group.

## Data Availability

No new data were created or analyzed in this study. Data sharing is not applicable to this article.

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
