# Peer review of "MRD-Based Therapeutic Decisions in Genetically Defined Subsets of Adolescents and Young Adult Philadelphia-Negative ALL"

_cancers, 2021, doi:10.3390/cancers13092108_

Round 1

Reviewer 1 Report

I have read this manuscript with plaesure and it gives a clear overview. 

I have no important remarks: I would only suggest that you maybe change the title a bit, as this is not only MRD (although indeed an impotant component) but also an overview of main genetic stratifying subjects, that in this world become more and more important (see ALLtogether protocol which is not only stratifying on MRD, but also on important genetic characteristics). 

This review is not only about MRD, so I would make the title broader. 

Minor comment: 

line 102: represneting ... should be representing

Author Response

enclosed file 

Reviewer 2 Report

This is a very thorough and comprehensive review on the treatment strategies in adolescent and young adult Philadelphia negative ALL. I have minor suggestions for modification:

  1. The authors shouldn’t use the term ”B-ALL”, which was used to describe leukemic phase of Burkitt’s lymphoma. B-cell precursor ALL should be preferably used.
  2. Lines 127/128 the concept of IKZF1plus ALL should be mentioned (Journal of Clinical Oncology 36, no. 12 (April 20, 2018) 1240-1249).
  3. The Figure 1 in fact is a table and should be treated like a table.
  4. Line 285 – the EuroFlow concept should be referred to Leukemia 2012 Sep;26(9):1908-75.
  5. Line 495 “MRDneg at high-risk” sounds strange.

Author Response

enclosed file
